# Phylogenetic Analysis and Protein Modelling of Isoflavonoid Synthase Highlights Key Catalytic Sites towards Realising New Bioengineering Endeavours

**DOI:** 10.3390/bioengineering9110609

**Published:** 2022-10-24

**Authors:** Moon Sajid, Shane R. Stone, Parwinder Kaur

**Affiliations:** UWA School of Agriculture and Environment, The University of Western Australia, Perth, WA 6009, Australia

**Keywords:** isoflavonoids, isoflavonoid synthase, heterologous biosynthesis, phylogenetic analysis, protein modelling, green modular bioindustries

## Abstract

Isoflavonoid synthase (IFS) is a critical enzyme for the biosynthesis of over 2400 isoflavonoids. Isoflavonoids are an important class of plant secondary metabolites that have a range of pharmaceutical and nutraceutical properties. With growing interest in isoflavonoids from both research and industrial perspectives, efforts are being forwarded to enhance isoflavonoid production in-planta and ex-planta; therefore, in-silico analysis and characterisation of available IFS protein sequences are needed. The present study is the first-ever attempt toward phylogenetic analysis and protein modelling of available IFS protein sequences. Phylogenetic analysis has shown that IFS amino acid sequences have 86.4% pairwise identity and 26.5% identical sites, and the sequences were grouped into six different clades. The presence of a β-hairpin and extra loop at catalytic sites of *Trifolium pratense*, *Beta vulgaris* and *Medicago truncatula*, respectively, compared with *Glycyrrhiza echinata* are critical structural differences that may affect catalytic function. Protein docking highlighted the preference of selected IFS for liquiritigenin compared with naringenin and has listed *T. pratense* as the most efficient candidate for heterologous biosynthesis of isoflavonoids. The in-silico characterisation of IFS represented in this study is vital in realising the new bioengineering endeavours and will help in the characterisation and selection of IFS candidate enzymes for heterologous biosynthesis of isoflavonoids.

## 1. Introduction

Isoflavonoid synthase (EC 1.14.14.87), a membrane-bound cytochrome P450 monooxygenase, is the key enzyme responsible for the biosynthesis of isoflavonoids [1,2]. Migration of the aryl ring from the C2 to C3 position is the first step toward synthesising the isoflavonoid core skeleton from flavonoid precursors (Figure 1) [3]. Being the first enzyme of the pathway, IFS directs the flow of intermediates from the flavonoid pathway to the isoflavonoid pathway and, therefore, defines a branch point in the synthesis of plant natural products [4]. Isoflavonoids are a unique sub-class of flavonoids that play an essential role in plant growth and development. In addition, the role of isoflavonoids in human health is well established, with recent studies highlighting their pharmacological properties ranging from hot flashes to cancer treatment [5,6,7,8]. It is envisaged that further characterisation will broaden the range of applications of isoflavonoids for human health.

Isoflavonoids are mainly identified from *Papilonoidae*, a sub-family of *Leguminosae*; however, up to 59 non-leguminous plant families (i.e., *Iridaceae*, *Rosaceae*, and *Liliaceae*) have also been reported to synthesise isoflavonoids [9]. As legumes are primary isoflavonoid producers, it was initially believed that the IFS gene is present in legumes only [10,11,12]. Increasing evidence of isoflavonoid’s presence in non-leguminous plants was followed by the isolation of the IFS gene from sugar beet, proving that IFS is not limited to legumes only [10]. Isoflavonoids play multiple roles in plant growth and development, and their role in establishing plant-microbial interactions has been well documented [13]. Over 2400 isoflavonoids have been identified from over 300 plants [3]. As IFS is responsible for synthesising parent isoflavonoids, the widespread presence of isoflavonoids indicates the importance and diversity of the enzyme [14,15]. It is assumed that the aryl migration activity of IFS must have emerged through a rare gain-of-function event by mutations from ancestral CYP93 genes and have survived and become dominant in present-day legumes due to the ecophysiological importance of the products it makes [16].

Advancements in synthetic biology and metabolic engineering made it possible to replicate the complete biosynthetic pathway of any natural plant product to heterologous hosts (specifically in microorganisms) [17]. Heterologous biosynthesis of isoflavonoids in microorganisms, i.e., *Saccharomyces cerevisiae*, and *Escherichia coli*, has been achieved (Table 1). Most of these studies highlighted low isoflavonoid titer due to low IFS expression, functionality, or incompatibility with a co-expressed CPR partner. A better understanding of the IFS sequence and structure relationship, along with the addition of more chassis (*Yarrowia lipolytica*), co-culturing as well as cell-free biosynthesis, are a few avenues that can help develop highly efficient heterologous isoflavonoid production systems [18,19,20]. In addition to this, applications of protein engineering and directed evolution strategies can be employed to engineer IFS isoforms capable of synthesising new-to-nature isoflavonoids with improved pharmacokinetics at industrially acceptable titer, rate and yield [5,21].

Isoflavonoids (biochanin-A, formononetin and genistein) are recently characterised as potential anti-cancer compounds, and demand for isoflavonoids is rising [3]. The market value of isoflavonoids was US $13.5 billion in 2018 and is expected to rise to US $30 billion by 2025 [30]. However, inefficient, traditional extraction methods, low plant yield, climate change, and priority to agricultural land use have questioned the availability of isoflavonoids to the general public [5,31]. Therefore, there is a drive for the biotechnology industry to produce isoflavonoids by using microorganisms sustainably and responsibly rather than using vital and valuable arable land [3]. Additionally, being a promising alternative, heterologous biosynthesis systems face low yield issues due to slow enzyme kinetics [23,27,28]. Therefore, the present study explores genetic variability in IFS protein sequences across different plant lineages and identifies key residues that define enzymes’ reactivity. Our results highlighted critical structural differences at catalytic sites that are assumed to play an important role in catalysis. Additionally, the preference of IFS for liquiritigenin (LQN) compared with naringenin (NGN) has been documented to help select the appropriate IFS isoform for heterologous isoflavonoid biosynthesis.

## 2. Materials and Methods

### 2.1. Sequence Retrieval

A BLASTP search was performed in the NCBI Non-redundant protein sequences (nr) database (https://blast.ncbi.nlm.nih.gov/Blast.cgi (accessed on 6 March 2022)) using *Glycyrrhiza echinata* IFS (One of the first identified and well-characterized IFS sequences) as the query sequence to retrieve available IFS protein sequences. Overall, 138 amino acid sequences representing 37 species having E-value = 0, and all key amino acids, i.e., Ser-310, Leu-371, Lys-375, and Arg-104 (amino acid numbers based on *G. echinata* sequence), were retrieved. P450 Engineering Database, University of Stuttgart (https://cyped.biocatnet.de/sequence-browser (accessed on 6 March 2022)), is also a valuable resource for the P450 family; however, it has fewer IFS entries as compared with NCBI. The detailed information (accession number etc.) about all the IFS sequences used in this study is listed in Appendix A.

### 2.2. Sequence Alignment and Phylogenetic Analysis

The 139 IFS amino acid sequences (138 retrieved from NCBI and *G. echinata* sequence) were aligned using the MUSCLE alignment algorithm with default parameters. A phylogenetic tree was constructed using the Neighbor-Joining algorithm with default parameters on Geneious Prime version 2021.1.1, and the results were documented.

### 2.3. Protein Modelling Analysis

Protein homology models of selected IFS sequences were developed on the Alphafold homology modelling server and by using *Salvia miltiorrhiza* CYP76AH1 (PDB: 5YLW) structure as a template on the SWISS-MODEL protein structure homology modelling server (https://swissmodel.expasy.org (accessed on 10 April 2022)). Protein homology models were analyzed in PyMOL.

### 2.4. Protein Docking Analysis

For protein docking, MGLTools v1.5.7 has been used. Briefly, target .pdb files were developed as discussed under the Section 2.3. 3D conformers of LQN (PubChem CID: 114829) and NGN (PubChem CID: 932) ligands files were downloaded from the PubChem database (https://pubchem.ncbi.nlm.nih.gov/ (accessed 21 May 2022)) in .sdf format and were converted to .pdb format using BIOVIA Discovery Studio Visualizer. Target and ligand .pdbqt files were prepared using AutoDockTools v4.2. In the docking simulation, the grid box size was set to 70 Å × 40 Å × 40 Å with a space of 0.375 Å in the structure of enzymes. The Lamarckian genetic algorithm was used to predict the binding modes of the target and ligands. The docking parameters were set for a population size of 150. The GA runs were set to 25, and the maximum number of generations was set at 25,000 for each docking study. The molecular docking was executed in Cygwin, and the results were analysed in AutoDockTools v4.2. Finally, the docked complex structure of the ligands and the surrounding amino acid residues were analysed in PyMOL and BIOVIA Discovery Studio Visualizer.

## 3. Results

### 3.1. Phylogenetic Analysis of IFS Amino Acid Sequences

Multiple sequence alignment has shown that the selected IFS amino acid sequence has 86.4 % pairwise identity and 26.5% identical sites (Figure 2). High sequence similarity among P450 family members has been reported, which is very clear from our results [15]. Following alignment, a phylogenetic tree was developed which grouped selected IFS amino acid sequences into at least 6 different clades (Figure 3, Appendix A).

#### 3.1.1. Clade 1

There were 8 IFS sequences in clade 1; they all belong to the genus *Lupinus* and represent three species (*L. albus*, *L. angustifolius*, and *L. luteus*). These sequences have 84.0% pairwise identity, 62.9% identical sites, and 59.2% similarity with the *G. echinata* IFS sequence (used as a template for comparison with selected IFS sequences). Sequences of this group show variations at substrate recognition sites 2 (SRS2) and SRS3 positions (Appendix A). These sequences have Glu at position 237 instead of Asp; however, both amino acids are acidic, so this substitution might not affect the enzyme’s catalytic efficiency. Additionally, they have Lys instead of Arg at position 112 in SRS1 compared to *G. echinata*. Members of this clade also have a deletion at 426 positions compared with the *G. echinata* IFS sequence.

#### 3.1.2. Clade 2

IFS amino acid sequences of *Cicer arietinum*, *Cullen corylifolium*, *Abrus precatorius*, *Lotus japonicus*, *Onobrychis viciifolia* and *Mucuna pruriens* are grouped in clade 2. These 10 sequences have 85.5% pairwise identity, 62.7% identical sites, and 61.6% similarity with the *G. echinata* sequence. *O. viciifolia* has an addition of 4 amino acids at positions 427–430 compared with *G. echinata* and other clade members.

#### 3.1.3. Clade 3

Twenty-one IFS sequences representing the genus *Glycyrrhiza*, *Arachis*, *Astragalus*, *Caragana*, *Vigna* and *Phaseolus* are grouped in clade 3. These sequences have 86.0% pairwise identity, 65.6% identical sites, and 65.2% similarity with the *G. echinata* sequence. The addition of Glu at position 304 in *P. vulgaris* compared with other members and *G. echinata* IFS is unique to this clade.

#### 3.1.4. Clade 4

IFS sequences from the genus *Pisum*, *Trifolium*, and *Medicago* are grouped in clade 4. These 19 IFS sequences have 89.6% pairwise identity, 67.9% identical sites, and 67.1% similarity with the *G. echinata* sequence. All four sequences of *P. sativum* and two sequences of *G. echinata* have the addition of Glu and Arg/Lys at positions 283–284 as compared with other clade members. Similarly, 3 of *M. truncatula* IFS seq. (QBF58773.1, QBF58774.1 and QBF58775.1) have the addition of Glu at position 326 compared with *G. echinata* and other clade members.

#### 3.1.5. Clade 5

Clade 5 has 12 sequences representing 3 genera (*Phaseolus*, *Vigna*, and *Pueraria*). These sequences have 94.1% pairwise identity, 84.9% identical sites, and 77.1% similarity with *G. echinata* sequence. All these sequences have a deletion of two amino acids at positions 259 and 260 compared with the *G. echinata* sequence. Together with that, the addition of Glu at position 302 is unique to *P. vulgaris* IFS sequence (QBF58778.1) compared with the *G. echinata* and other clade members.

#### 3.1.6. Clade 6

Sixty-nine IFS sequences representing 10 genera (*Cajanus*, *Lens*, *Trifolium*, *Pisum*, *Lupinus*, *Beta*, *Vigna*, *Medicago*, *Vicia* and *Glycine*) are grouped in clade 6. It is clear from Figure 3 that clade 6 is dominated by *G. max* and *G. soja* IFS sequences. These 69 sequences have 96.6% pairwise identity, 53.0% identical sites, and 51.1% similarity with the *G. echinata* template. *G. max* IFS seq. AAT47734.1 has the addition of 5 amino acids at position 232-236 and *G. max* IFS seq. QBF58777.1 and QBF58777.1 have a Glu addition at position 302 compared with the *G. echinata* and other clade members.

### 3.2. Protein Homology Modelling of Selected IFS Candidates

Being a P450 class enzyme, instability, scarcity, and the presence of many homologous proteins make solubilisation and purification challenging; that is why the crystal structure of IFS has not been solved yet [1]. However, a few key amino acids; Ser-310, Leu-371, Lys-375, and Arg-104 (amino acid numbers based on *G. echinata* sequence) involved in substrate selection and aryl ring migration have been identified (Figure 1) [14,32]. Based on these findings, it was concluded that the substitution of Ser-310 could suppress; however, Lys-375 is critical for the aryl migration activity of IFS, and Arg-104 is essential to maintain P450 architecture [14]. Many IFS encoding genes have been identified and characterised from various legumes and non-legume plants (Appendix A). Following phylogenetic analysis, 38 representative IFS sequences (at least 1 IFS seq./species and 2 for *G. max*) were selected for homology modelling (Appendix A). Protein models developed on the SWISS-MODEL server have shown sequence identity ≥29.36% and coverage ≥ 0.82. For a few sequences, more than 2 models were developed, and a model having a higher GMGQ score was selected for further analysis (Appendix A). On the other hand, protein models developed on Alphafold having the lowest E-value were selected for further analysis (Appendix A).

Protein homology modelling has highlighted structural variations at the catalytic site for *Astragalus mongholicus* (AEH68209.1), *Beta vulgaris* (AAF34538.1), *Lens culinaris* (AAF34525.1), *Medicago truncatula* (AAO16603.1), *Trifolium pratense* (AAP06953.1) and *Vigna angularis* (XP_017424255.1) compared with *G. echinata* (BAA76380.1) (Appendix A). Following that, we developed homology models of the remaining IFS isoforms of these species; *Beta vulgaris*: AAF34537.1, *Lens culinaris*: AAF34526.1, *Medicago truncatula*: KEH30959.1, KEH30962.1, QBF58773.1, QBF58774.1, QBF58775.1, RDX63227.1, *Trifolium pratense*: XP_045826491.1 and *Vigna angularis*: XP_017422719.1 (Appendix A). However, a significant structural difference was only seen for *B. vulgaris* (AAF34538.1), *T. pratense* (AAP06953.1) and *M. truncatula* (QBF58774.1) IFS homology models compared with the *G. echinata* (BAA76380.1). On the other hand, protein models developed on the Alphafold server have not shown any extra β-hairpin or loop structure; instead, a new β-sheet (β4-sheet) has been seen in all models. β4-sheet is believed to be an integral part of the P450 structure and was also proposed for IFS by Sawada et al.; however, the β4-sheet is not present in the crystal structure of *Salvia miltiorrhiza* CYP76AH1 (PDB: 5YLW), which is quite surprising [14,32,33]. Therefore, the presence of β4-sheet can only be confirmed by solving the 3D model of IFS.

The presence of an extra β-hairpin near the active site in *B. vulgaris* (AAF34538.1) and *T. pratense* (AAP06953.1) and a long loop in *M. truncatula* (QBF58774.1) compared with the *G. echinata* (BAA76380.1) protein model is a significant difference in protein structures (Figure 4). The extra β-hairpin and loop structure are present over the heme unit, inside the active site and are surrounded by key amino acids. The orientation of Leu-371 and Lys-375 has been changed due to the extra β-hairpin, and these amino acids are tilted more towards the heme unit and Ser-310 (Appendix A). On the other hand, the orientation of Ser-310, Leu-371 and Lys-375 in models of *B. vulgaris* (AAF34538.1), *T. pratense* (AAP06953.1), *M. truncatula* (QBF58774.1) and *G. echinata* (BAA76380.1) developed on Alphafold are different compared with models developed on SWISS-MODEL (Figure 4). This change in the position of key amino acids might affect the enzyme’s catalytic efficiency.

Similarly, the effect of addition and deletion of amino acids, i.e., ER at position 260–261 in clade 4–6 (amino acid numbers are based on *G. echinata* sequence), was studied. The addition and deletion of amino acids are away from the active site, in loops structure present over the protein surface. These structural changes, along with the substitution of amino acids in SRS sequences, may affect substrate binding and interaction with CPR partners. Therefore, analysis of the effects of these variations is a potential area of research.

The amino acid sequence ‘KVSMEERAGLTVP’ (*G. echinata* amino acid 489–501) of *G. echinata*, *B. vulgaris*, *M. truncatula* and *T. pratense* at the extra β-hairpin site, and loop site was unchanged except for only two amino acids substitutions present in *T. pratense* (AAP06953.1) sequence (Appendix A). So, to check whether the extra β-hairpin are artefacts generated during the homology modelling, *B. vulgaris*, *M. truncatula*, *T. pratense* and *G. echinata* protein models were aligned with *S. miltiorrhiza* CYP76AH1 (PDB: 5YLW) structure in PyMOL (Appendix A). Key amino acid sites, I-helix, and all other structures except for the long loop of *M. truncatula* and extra β-hairpin of *B. vulgaris* and *T. pratense* were perfectly matched with the *S. miltiorrhiza* CYP76AH1 template. Additionally, the loop of *M. truncatula* and β-hairpin of *B. vulgaris* and *T. pratense* was not observed in any other IFS models. Thus, we concluded that the extra β-hairpin and loop might result from the addition and/or substitution of amino acids at other sites; therefore, further structural, and functional characterization of IFS is required.

Protein structure prediction models are becoming increasingly important in understanding sequence-structural relationships for proteins, like IFS, that do not have crystal structure solved [34,35]. Homology modelling is considered the most accurate of the available methods, and backbone prediction with sub-Angstrom Root Mean Square Distance (RMSD-Cα) has become a routine practice, and prediction of side chains has become increasingly accurate [34,35]. Protein models developed through homology modelling, even for challenging proteins, are now considered reliable [35]. Advancements in protein structure modelling have great potential because more reliable and accurate protein models will help better understand sequence-structure relationships.

### 3.3. Protein Docking of Selected IFS Candidates

Following protein modelling, the selected IFS; *B. vulgaris* (AAF34538.1), *T. pratense* (AAP06953.1), *M. truncatula* (QBF58774.1) and *G. echinata* (BAA76380.1) was docked with LQN and NGN as ligands (Figure 5, Table 2). The binding energy (ΔG) comparison highlighted the preference of IFS toward respective substrates [36,37]. All four IFS models have shown the lowest binding energy for LQN compared with NGN (Table 2). The difference in binding energy for LQN clearly explains the preference of IFS toward LQN in natural and heterologous hosts, as stated in the literature [38,39]. Protein docking analysis of IFS models developed on the SWISS-MODEL server has shown slightly lower binding energies compared with models developed using Alphafold (Table 2).

Docking results have highlighted *T. pratense* as the most efficient candidate for conversion of LQN to isoflavonoids compared with other IFS models. On the other hand, the binding energies of *T. pratense*, *M. truncatula* and *G. echinata* for NGN were almost the same. These IFS may show similar conversion efficiency in heterologous hosts. As there is not much difference in binding energies, the interaction of ligand (LQN) with the enzyme at the catalytic site can help us explain and select an IFS candidate. Docking analysis has shown that LQN develops interaction with the Fe-O complex at the C3 position of LQN (for hydrogen abstraction (Figure 1)) to start the catalysis reaction. The distance between the Fe-O complex and C3 of LQN is 3Å for *T. pratense*, which is increased to 3.4Å for *G. echinata* (Figure 5). The Fe-O and C3 of LQN interaction are not seen in any of the 25 docked complexes of *M. truncatula* and *B. vulgaris*. It is believed that LQN would develop interaction with Fe-O; however, further refinement of models using molecular simulation algorithms can help find that interaction. Chemler et al. have compared 5 IFS, including *T. pratense*, *G. echinata*, and *M. truncatula*, in a yeast system, and they have ranked *T. pratanse* as the most efficient candidate for bioconversion of NGN [40]. Compared with other models, the low binding energy of *T. pratense* for LQN and NGN makes it the most efficient candidate for the biosynthesis of isoflavonoids in heterologous hosts (Table 1).

Docking results have highlighted a few more key amino acids that might play an important role in catalysis/aryl migration reaction and in stabilising the substrate in the active site over the heme unit. Interaction of the 7-OH group with Val-373 and 4′-OH with Gly-307, Thr-308 and Asp-309 of LQN (amino acid number based on *G. echinata* sequence) are significant findings. Val-119, Ala-120 and Met-121 as well as Arg-376, seem to be involved in heme stabilization. Previously, the interaction of Leu-371 with the O atom of the C-ring of LQN based on a homology model of *GeIFS* built using the crystal structure of bacterial BM3 as a template was reported; however, the authors concluded that Leu-371, along with Ser-310 and Lys-375 (amino acid number based on *G. echinata* sequence) specifically play a role in accommodation of substate and stabilisation of P450 architecture [14,32]. Together with that, the catalytic importance of the extended ring present over the catalytic site in the homology model of *M. truncatula* and extra β-hairpin in *B. vulgaris* and *T. pratense* have been highlighted by docking results. The orientation of ligands has been slightly changed in *M. truncatula* due to amino acid residues in the loop. In addition, the change in orientation of Lys-375 due to β-hairpin in *B. vulgaris* and *T. pratense* may be the reason for the higher catalysis rate of *T. pratense* [40]. However, in-depth protein docking analysis with more models/samples is required.

## 4. Discussion

In the present study, phylogenetic analysis and protein modelling has been performed on selected IFS amino acid sequences representing 37 species. Phylogenetic analysis has shown 86.4% pairwise identity and 26.5% identical sites among IFS amino acid sequences, highlighting a recent origin of IFS (CYP93C subfamily) from the CYP93B subfamily [15]. It is believed that CYP93C and CYP93B families share the same origin, further supported by the evidence that these subfamilies share the same substrate [41,42]. As subfamily CYP93C originated from CYP93B, Sawada et al. replaced key amino acids (Ser-310 and Lys-375) in the CYP93B subfamily enzyme; however, no aryl migration activity was seen [14,32]. The authors conclude that the key amino acids (Ser-310, Lys-375 and Leu-371, Arg-104) identified through mutational studies are not solely responsible for aryl-migration activity [14]. It was proposed that further amino acids in the IFS enzyme having a role in aryl migration reaction are assumed to be present, and overall protein architecture is an important factor that controls aryl-migration activity [14]. Together with that, the formation of the by-product (3-hydroxyflavone) also indicates the inefficiency of the aryl-migration apparatus, which means the answer to IFS enzymatic efficiency lies in amino acids sequence variability, and screening of different IFS isoforms can help to select highly efficient IFS candidate. In this regard, variations in amino acid sequences highlighted in the present study can help direct future studies for better IFS structure-function characterisation. It is envisaged that substitutions at distal sites might affect enzyme reactivity differently [43]. Therefore, further studies are required to better understand the function of amino acids identified in the present study.

IFS, a P450 enzyme, requires electrons from its partner enzyme, known as cytochrome P450 reductase (CPR), for catalysis [44]. That means IFS interacts with CPR and substrate molecules simultaneously, and anything that affects the stability of this interaction will influence enzyme reactivity and/or rate of catalysis. Literature indicates that the source of IFS and CPR strongly influence the rate of isoflavonoid biosynthesis [40]. That means amino acids involved in the electron transfer pathway present in CPR and IFS may have a preference for each other, and the overall architecture of both enzymes may have a role in electron transportation [14,40]. Therefore, amino acids that play a role in electron transfer and in, substrate selection, stabilisation and interaction with the heme unit and are involved in the recruitment of other enzymes require further insights [43,45,46]. Recent developments in molecular simulations can help advance our understanding specifically of IFS-CPR interaction, electron transfer pathway and substrate entrance channels [47]. Such studies will further strengthen our knowledge about aryl ring migration reaction and catalytic efficiency, which will help select appropriate IFS candidates for heterologous isoflavonoid biosynthesis.

With recent developments in synthetic biology and metabolic engineering, there is a drive for the biotechnology industry to enhance isoflavonoid production in-planta and ex-planta [48,49]. The potential to introduce exogenous isoflavonoid biosynthesis into cereal crops is an exciting avenue, but one in which the outcomes may be limited due to the negative public sentiment and regulatory barriers surrounding genetically modified crops [31]. The proof of concept evidence in onion and rice demonstrates that the approach is viable and likely beneficial to the agricultural industry [50,51]. While there is an excellent attraction toward the biofortification of crops through genetic engineering, societal perceptions and regulatory barriers will need to be addressed and overcome for this approach to be of value. On the other hand, microorganisms are a more attractive tool towards the generation of isoflavonoids due to their scalability, small footprint and rapid production [52]. As isoflavonoid products would be either produced through generally regarded as safe microorganisms, i.e., yeast or isolated and purified from the growth media, regulatory hurdles would be less restrictive [18]. Additionally, developing prokaryotic compatible IFS for scaling in bacterial systems has immense potential; however, this requires a large amount of research work, including protein engineering, to make IFS a cytosolic self-sufficient enzyme like prokaryotic P450 [44,53].

In conclusion, the structure-function characterisation of IFS for the highest isoflavonoid biosynthesis in plants and microorganisms, specifically in prokaryotic systems, has immense potential to transform future farming systems and the biotechnology industry.

## Figures and Tables

**Figure 1 bioengineering-09-00609-f001:**
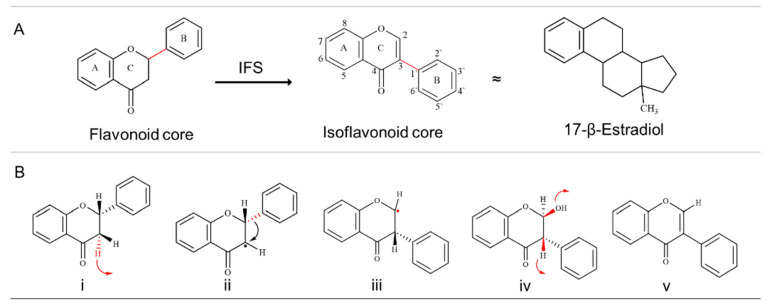
Basic structure of isoflavonoid and aryl ring migration reaction mechanism. (**A**) B-ring is attached at the C3 position in isoflavonoids, making them different from flavonoids and similar to the human estrogen hormone (17-β-Estradiol). (**B**) The aryl ring migration reaction can be divided into five steps (Red colour represents the site of reactivity). (i) A H atom from the C3 position is removed (red arrow represents the removal of H), (ii) which is followed by the migration of the aryl ring from C2 to the C3 position. (iii) Due to aryl ring migration, C2 becomes electron-deficient, which is (iv) stabilised with the addition of the OH group, (v) A dehydration step that either occurs spontaneously or with the help of 2-hydroxyisoflavonid dehydratase (HIDH) completes the reaction. Step i to iv are catalysed by IFS. IFS: Isoflavonoid Synthase.

**Figure 2 bioengineering-09-00609-f002:**
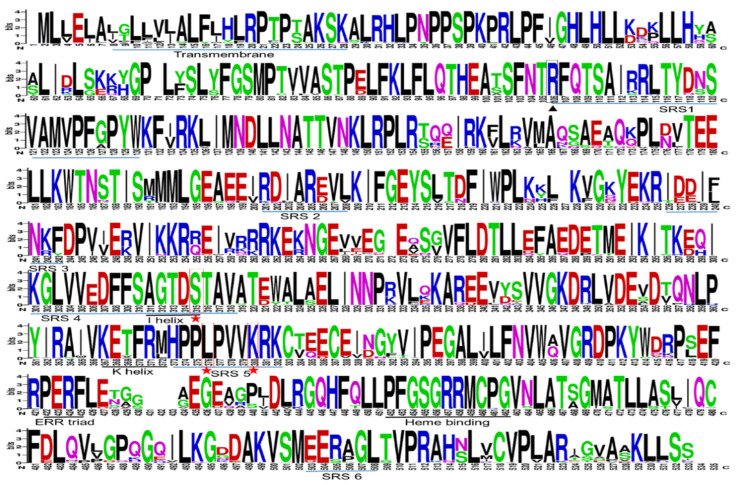
Sequence logos of the multiple sequence alignments of the 139 IFS amino acids sequences: The sequence logos are based on IFS (CYP93C subfamily) amino acids alignment using the Neighbor-Joining algorithm on Geneious Prime. The logos were generated using Weblogo (https://weblogo.berkeley.edu/logo.cgi (accessed on 26 March 2022)). The bit score indicates the information content for each position in the sequence. Hight of the letters designating the amino acid residues at each position represents the degree of conservation. The key conserved motifs are underlined; the black line indicates the P450 motifs, and the blue ones indicate substrate recognition sites (SRSs). Red stars and boxes represent key amino acids, and black triangles and boxes represent important amino acids that also have an important role in catalysis.

**Figure 3 bioengineering-09-00609-f003:**
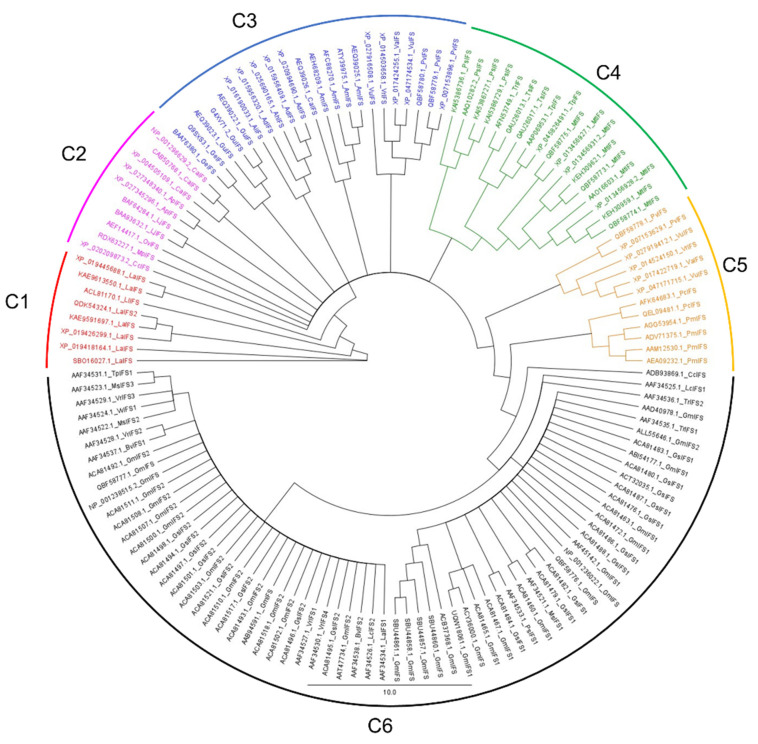
Phylogenetic tree of IFS amino acid sequences. The phylogenetic tree is constructed using the Neighbour Joining (NJ) algorithm for 139 IFS amino acid sequences. Branch labels represent the NCBI accession number followed by species name initials and IFS isoform number. Colours are used to differentiate between clades. The scale bar represents amino acid substitutions per site.

**Figure 4 bioengineering-09-00609-f004:**
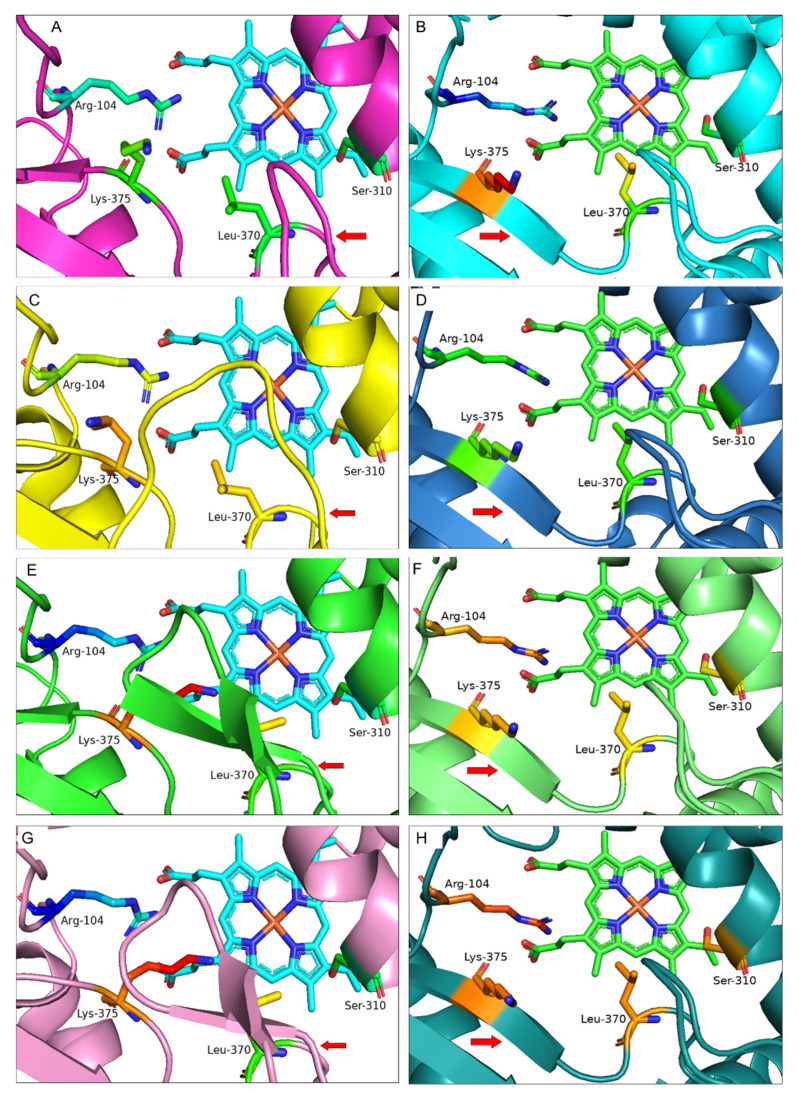
Active site structure of IFS from (**A**,**B**) *G. echinata* (BAA76380.1) (**C**,**D**) *M. truncatula* (QBF58774.1) (**E**,**F**) *B. vulgaris* (AAF34538.1) and (**G**,**H**) *T. pratense* (AAP06953.1). Protein models on the left (**A**,**C**,**E**,**G**) are developed using *Salvia miltiorrhiza* CYP76AH1 (PDB: 5YLW) as a template on the SWISS-MODEL protein structure homology modelling server (https://swissmodel.expasy.org (accessed on 10 April 2022)). Protein models on the right (**B**,**D**,**F**,**H**) are developed on the Alphafold server. Key amino acids are shown as sticks; amino acid numbering is based on the *G. echinata* amino acid sequence. Red arrows indicate the presence of a long loop in *M. truncatula* and extra β-hairpin of *B. vulgaris* and *T. pratense* compared with the *G. echinata* protein model and the β4-sheet present in all models. The presence of an extra β-hairpin in the active site over the Heme unit might control/interfere with the entrance and selection of substrate and may have a role in enzyme efficiency.

**Figure 5 bioengineering-09-00609-f005:**
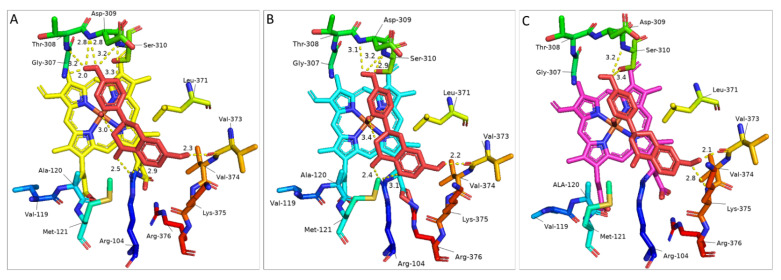
Key amino acids along with Liquiritigenin (docked as a ligand) in selected IFS homology models: (**A**) *T. pratense* (AAP06953.1), (**B**) *G. echinata* (BAA76380.1) and (**C**) *M. truncatula* (QBF58774.1). Key amino acids and LQN structure are shown in the stick model. LQN is shown in red; key amino acids are shown in rainbow colours. Polar interactions are shown in yellow dotted lines, and the distance is shown in Å units.

**Table 1 bioengineering-09-00609-t001:** Heterologous expression of IFS in microorganisms.

Species	Heterologous Host	Substrate (Input)	Isoflavonoid (Product)	Genetic Engineering Strategy	Ref.
*Glycine max* IFS *Medicago sativa* CHI	Yeast	IsoLQN/LQN	DEN	Construction of bi-functional enzyme by in-frame gene fusion for higher yield	[22]
*G. max* IFS Poplar hybrid CPR	Yeast	Tyrosine, *p*-coumaric acid	0.1–7.7 mg/L GEN	Metabolic engineering of yeast using plasmid-based gene expression for heterologous biosynthesis	[23]
*Glycyrrhiza echinata* IFS	Yeast/ *E. coli*	3 mM Tyrosine	6 mg/L GEN	Co-culture approach for better expression and higher yield	[24]
*G. max* IFS *Catharanthus roseus* CPR	*E. coli*	0.05 mM NGN	GEN	Engineered IFS architecture to complement a self-sufficient bacterial P450 enzyme	[25]
		0.05 mM LQN	DEN		
*G. max* IFS *Putina hybrdia* CPR	Yeast	10 mM Phenylalanine	0.1 mg/L GEN	Functional expression of plant enzyme and construction of pathway in yeast chassis	[26]
		1 mM p-Coumaric acid	0.14 mg/L GEN		
		0.5 mM NGN	7.7 mg/L GEN		
*Trifolium pretense* IFS *Oryza sativa* CPR	*E. coli*	500 μM NGN	35 mg/L GEN	Engineering of IFS for expression in prokaryotic system. Optimisation of culture system, medium, growth conditions and substrate concentration to increase overall yield	[27]
		300 μM p-Coumaric acid	18.9 mg/L GEN		
*G. echinata* IFS *C. roseus* CPR	Yeast	-	9.9 mg/L DEN	Metabolic engineering of a yeast strain for de-novo isoflavonoids biosynthesis	[28]
*Lotus japonicas* IFS *L. japonicas* CPR	Yeast	-	19.32 mg/L GEN	Modular engineering of yeast and screening of IFS for de-novo biosynthesis of genistein	[29]

Abriviations: IsoLQN: Isoliquiritigenin, LQN: Liquiritigenin, NGN: Naringenin, DEN: Daidzein, GEN: Genistein.

**Table 2 bioengineering-09-00609-t002:** Binding energy (ΔG) comparison of selected IFS docked with LQN and NGN.

Species	ΔG for LQN		ΔG for NGN	
	Swiss-Model	Alphafold	Swiss-Model	Alphafold
*B. vulgaris*	−8.03	−7.68	−7.80	−7.18
*G. echinata*	−7.37	−7.98	−6.97	−7.49
*M. truncatula*	−7.46	−7.76	−7.12	−7.51
*T. pratense*	−8.23	−7.98	−7.83	−7.50

## Data Availability

Not applicable.

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
