# Peer review of "Phylogenetic Analysis and Protein Modelling of Isoflavonoid Synthase Highlights Key Catalytic Sites towards Realising New Bioengineering Endeavours"

_bioengineering, 2022, doi:10.3390/bioengineering9110609_

Round 1
Reviewer 1 Report
The manuscript entitled “Phylogenetic analysis and protein modelling of isoflavonoid synthase highlights key catalytic sites towards realising new bioengineering endeavours “ can be of interest to wide readers of journals and contributes to existing knowledge on the subject matter. However, I have pointed out few pertinent points for improving the clarity of the content and boosting the scientific soundness of the manuscript.
Abstract: Authors have not given due consideration to briefness but comprehensiveness requirement of the abstract. Include main findings of the research work.
Introduction
Line 27-28, 30-32, 56-58: Give citations
Line 61: Define “CYP93C enzymes”
Table 1: Write the full scientificfic names
Results
Adjust the citations used in the “results” section into the “discussions” section
Rewrite the conclusion. Need to present concrete findings based on recorded data,
Adjust figure 1 and Table 1 in the result section
Author Response
Reviewer #1
The manuscript entitled "Phylogenetic analysis and protein modelling of isoflavonoid synthase highlight key catalytic sites towards realising new bioengineering endeavours "can be of interest to wide readers of journals and contributes to existing knowledge on the subject matter. However, I have Commented on a few pertinent Comments to improve the clarity of the content and boost the scientific soundness of the manuscript.
Comment #1: Abstract:
The authors have not given due consideration to the briefness but comprehensiveness requirement of the abstract. Include the main findings of the research work
Answer: We appreciate this feedback and agree that highlighting the inclusion of findings in the abstract will help increase its effectiveness.
The text in black is replaced with red in the abstract:
Phylogenetic analysis has shown that IFS protein sequences can be grouped into six clades. Homology modelling has highlighted critical structural differences at catalytic sites for Glycyrrhiza echinata, Beta vulgaris, Medicago truncatula, and Trifolium pratense IFS. Phylogenetic analysis has shown that IFS amino acid sequences have 86.4 % pairwise identity and 26.5% identical sites, and the sequences were grouped into six different clades. The presence of a β-hairpin and extra loop at catalytic sites of Trifolium pratense, Beta vulgaris and Medicago truncatula, respectively, compared with Glycyrrhiza echinata, are critical structural differences that may affect catalytic function.

Reviewer 2 Report
The manuscript "Phylogenetic analysis and protein modelling of isoflavonoid synthase highlights key catalytic sites towards realising new bioengineering endeavours" by Moon Sajid et al. presents a phylogenetic analysis of isoflavonoid synthase (IFS) enzymes, followed by homology modeling and docking of two flavanones. In principle, the observations made could assist selection of candidate IFS enzymes for heterologous biosynthesis of isoflavonoids.
Considering that the main findings of this work are based on the homology models of the IFS structures, my main concern is actually addressed by the authors the last paragraph of section 3.2 (lines 259-267). As stated: "Protein models developed through homology modelling, even for challenging proteins, are now considered reliable [27]". So why did the authors employ homology models based on single X-ray structure (CYP76AH1 from PDB ID: 5YLW) using Swiss Model server, instead of using the most recent and reliable method of AlphaFold? ref. [27]
I would be glad to review a revised version of this work in which the authors employ models of AlphaFold to draw their conclusions. A comparison with the observations made using the homology models would be also a nice addition. Other than this, some minor issues are:
1. The introduction could be more comprehensive in providing bioengineering endeavors that substantiate the significance of this work.
2. Binding free energies (ΔG in kcal/mol) from AutoDock are estimates and should be considered tentative when comparing values that differ below the standard error of ~2.5 kcal/mol.
3. Figure 5 is not informative as it does not illustrate the protein-ligand interactions. Either use schematic LigPlot diagrams, or indicate residue-specific interactions in a more clear way (color-coded dashed / dotted lines with distances and larger fonts).
Based on the above, I cannot suggest publication of this manuscript to Bioengineering at its present form.
Author Response
We are grateful for the constructive feedback and suggestions to improve the MS. Please see the detailed response to reviewer's comments in the document attached.

Round 2
Reviewer 1 Report
The author of the manuscript has revised the MS, and responded all the comments and questions carefully. Now the MS can be accepted in its present form.
Reviewer 2 Report
The revised version of the manuscript has addressed most of my concerns adequately. Therefore the manuscript can be accepted for publication to Bioengineering.